# Rapid Visual Detection of Methicillin-Resistant *Staphylococcus aureus* in Human Clinical Samples via Closed LAMP Assay Targeting *mecA* and *spa* Genes

**DOI:** 10.3390/microorganisms12010157

**Published:** 2024-01-12

**Authors:** Noora S. A. Abusheraida, Asraa A. H. AlBaker, Asmaa S. A. Aljabri, Hana A. Abdelrahman, Hassan Al-Mana, Godwin J. Wilson, Khalid A. Anan, Nahla O. Eltai

**Affiliations:** 1College of Health Science, Qatar University, Doha P.O. Box 2713, Qatar; na1601370@student.qu.edu.qa (N.S.A.A.); aa1605885@student.qu.edu.qa (A.A.H.A.); aa2002182@student.qu.edu.qa (A.S.A.A.); 2Biomedical Research Center, Qatar University, Doha P.O. Box 2713, Qatar; hana.abdelrahman@qu.edu.qa (H.A.A.); h.almana@qu.edu.qa (H.A.-M.); 3Laboratory Medicine and Pathology, Hamad General Hospital, Doha P.O. Box 3050, Qatar; gwilson@hamad.qa; 4Ministry of Municipality, Doha P.O. Box 35081, Qatar; khalid.enan@gmail.com

**Keywords:** LAMP, MRSA, MSSA, HNB, visual/colorimetric dye

## Abstract

The emergence of antimicrobial resistance (AMR), particularly methicillin-resistant *Staphylococcus aureus* (MRSA), poses a significant global health threat as these bacteria increasingly become resistant to the most available therapeutic options. Thus, developing an efficient approach to rapidly screen MRSA directly from clinical specimens has become vital. In this study, we establish a closed-tube loop-mediated isothermal amplification (LAMP) method incorporating hydroxy-naphthol blue (HNB) colorimetric dye assay to directly detect MRSA from clinical samples based on the presence of *mecA* and *spa* genes. In total, 125 preidentified *S. aureus* isolates and 93 clinical samples containing *S. aureus* were sourced from the microbiology laboratory at Hamad General Hospital (HGH). The sensitivity, specificity, positive predictive value (PPV), and negative predictive value (NPV) were computed based on conventional PCR. The assay demonstrated 100% specificity, 91.23% sensitivity, 0.90 Cohen Kappa (CK), 100% PPV, and 87.8% NPV for the clinical samples, while clinical isolates exhibited 100% specificity, 97% sensitivity, 0.926 CK, 100% PPV, and 88.89% NPV. Compared to cefoxitin disk diffusion, LAMP provided 100% specificity and sensitivity, 1.00 CK, and 100% for PPV and NPV. The study revealed that the closed-tube LAMP incorporating (HNB) dye is a rapid technique with a turnaround time of less than 1 h and high specificity and sensitivity.

## 1. Introduction

Throughout history, infectious diseases have shaped humanity, creating a higher burden on health and the economy [1]. Over centuries, inspecting the pathogenicity mechanism of microorganisms in correlation with disease causing aided in developing antimicrobial drugs that reduce the spread of infectious diseases [2]. Antimicrobial resistance (AMR) is a critical and emerging global health threat that poses challenges to effectively preventing and treating infections caused by bacteria, viruses, parasites, and fungi [3,4,5,6,7]. Therefore, it is crucial to induce a higher spread of antimicrobial-resistant pathogens that restrain the ability to treat infectious diseases, particularly the global spread of multi- and pan-resistant bacteria known as superbugs, including methicillin-resistant *Staphylococcus aureus* (MRSA) [8]. *Staphylococcus aureus* (*S. aureus*) is a Gram-positive bacteria that often colonizes the human skin, mucous membranes, nose, and other areas without causing symptoms and is considered part of the normal flora [4]. Approximately 30% of the human population is colonized by *S. aureus* [9]. Over the past few decades, *S. aureus* has emerged with a new form resistant to a wide range of beta-lactam antibiotics, including penicillin, methicillin, amoxicillin, nafcillin, oxacillin, and cephalosporins known as MRSA [10]. The ability of MRSA to produce a protein called penicillin-binding protein (PBP2a) led to its resistance to multiple beta-lactam antibiotics; the protein allows the bacteria to escape the antibiotic inhibitory effect [11,12]. This means that healthcare providers are facing a new challenge when treating MRSA clinical manifestations. 

Identifying MRSA and methicillin-sensitive *Staphylococcus aureus* (MSSA) ranges from various cultures to molecular methods [13,14]. These include chromogenic agar, latex agglutination, matrix-assisted laser desorption/ionization-time of flight MALDI-TOF, biochemical identification, and Vitek susceptibility testing. The run-time for these assays ranged between 3 min and 12 h and 48 h for chromogenic agar. However, they all require bacteria culture and organism isolation first, which adds at least 16 h to the total turnaround time. On the other hand, nucleic acid-based reaction methods are widely implemented and based commonly on *detecting S. aureus*-specific genes targeting the *mecA* gene or *mecC* [13]. These molecular methods include end-point polymerase chain reaction (PCR), and various molecular automated detection systems showed sensitivity to clinical specimens that ranged from 69.2 to 100% and specificity from 64.5 to 100% with a turnaround time that fluctuated from 1 h to 3 h [15]. The most significant barrier when considering molecular assays is the cost. Thus, designing a reasonable strategy for rapidly screening MRSA directly from clinical samples is crucial. In the last decade, loop-mediated isothermal amplification (LAMP) was reported as a novel nucleic acid amplification method [16]. It is applied to detect various pathogenic organisms [16,17,18]. LAMP provides the shortest run-time, high specificity, and sensitivity ranging from 91.3 to 98.4%. Unlike PCR, LAMP is performed under isothermal conditions without requiring a thermal cycler [19].

Most laboratories choose between chromogenic media and real-time PCR, which can produce results within 24 h [20]. However, chromogenic media has a sensitivity of about 80% at 25 h, and it must be incubated longer, up to 48 h, for the sensitivity to approach 100% [20]. Highly efficient screening protocols and rapid implementation of infection control practices, together with the proper treatment, are crucial to controlling and limiting the nosocomial spread of MRSA [21]. Accurately detecting *mecA*-mediated ß-lactam resistance in S. aureus is essential for treating overt infections [22]. For this reason, this study aims to develop an alternative and rapid MRSA detection method that can be adopted at diagnostic laboratories and hospitals in Qatar. This can be achieved by implementing closed-tube LAMP with the colorimetric dye hydroxy-naphthol blue (HNB), which can detect *mecA* and *spa* genes. HNB serves as a metal ion indicator, eliminating the need for intercalating dyes such as SYBR green, which can intensify the occurrence of aerosol contamination when tubes are opened for additions [23]. The majority of non-specific detection and the potential for misleading false positive/negative results in LAMP reactions are primarily attributed to cross-contamination, which can occur due to the chance of both cis and trans priming of the LAMP primers [24]. Additionally, the study aims to compare the efficacy of a closed-tube LAMP assay to conventional PCR and disk diffusion tests and determine its detection limit. 

## 2. Materials and Methods

### 2.1. Clinical Specimens and S. aureus Isolates

Two hundred and eighteen samples were obtained from Hamad General Hospital (HGH), Doha, Qatar, and collected from patients suspected of having bacterial infections. The samples were identified using MALDI-TOF (Bruker Daltonik GmbH, Bremen, Germany) as per manufacturer protocol. These samples are categorized into two groups: clinical specimens, consisting of 57 MRSA and 36 MSSA, and bacterial isolates, comprising 101 MRSA and 24 MSSA. The clinical specimens, such as blood and tissue, were collected from patients suspected of having *S. aureus* infections. Clinical isolates were obtained from confirmed specimens of *S. aureus*-infected patients through laboratory culture. The study was conducted in full conformance with principles of the “Declaration of Helsinki”, Good Clinical Practice (GCP), and within the laws and regulations of the Ministry Of Public Health (MoPH) in Qatar, following the acquisition of Institutional Biosafety Committees (IBC) (Doha, Qatar), QU-BRC-2021/050, and MRC-01-20-1216 from Hamad Medical Corporation (HMC) (Doha, Qatar). The clinical specimens and isolates with confirmed MRSA and MSSA were dispatched to the microbiology laboratory at the Biomedical Research Center, Qatar University, to evaluate closed-LAMP using HNB colorimetric dye as a diagnostic technique without any patient information. Clinical specimens were processed within 24 h of receipt, while clinical isolates were previously collected and stored at −80 °C until further analysis. Quality control organisms utilized in this study included *E. coli* ATCC 25922 as a negative control for *mecA* and *spa* genes, *S. epidermidis* ATCC 12228 as a positive control for *mecA* gene, MRSA ATCC BAA-976 as a positive control for both genes, and MSSA S15 as a positive control for the *spa* gene. All ATCC control strains were collected from the American Type Culture Collection, Manassas, Virginia, USA, and MSSA S15 is an identified strain from Hamad Medical Corporation (Doha, Qatar). 

### 2.2. Cefoxitin Disk Diffusion Method (Kirby–Bauer Test)

One hundred and twenty-five *S. aureus* clinical isolates were tested for antibiotic susceptibility using the standard disk diffusion technique as recommended by Clinical and Laboratory Standards Institute CLSI guidelines [25]. A single colony of an overnight culture of the isolates was suspended in phosphate-buffered saline (PBS, Atom Scientific, Hyde, UK) to achieve an inoculum of 0.5 McFarland as measured by DensiCHEK Plus (bioM’erieux, Marcy l’Etoile, France). Suspensions were fully swabbed on Mueller–Hinton agar plates (Himedia, Mumbai, India). Then, a 30 µg cefoxitin disk (Liofilchem^®^, Roseto Degli Abruzzi, Italy) was applied to the agar surface using sterile forceps, and plates were incubated at 37 °C for 18 to 24 h. The zone of inhibition was measured to determine the Minimum Inhibitory Concentration (MIC) and interpreted according to the CLSI guidelines [25].

### 2.3. DNA Extraction 

DNA was extracted from an overnight culture of clinical isolates. A few single colonies were suspended in phosphate-buffered saline (PBS, Atom Scientific, Hyde, UK) to prepare a bacterial suspension. The bacterial suspension was then incubated in a Heat Block PCMT Thermo-shaker (Grant Bio, Cambridge, UK) for 10 min under 100 °C and subjected to centrifugation for 5 min at 2000 rpm. Also, DNA was extracted from the clinical specimens, like swab and tissue, once received by boiling and proceeded as above.

### 2.4. Closed-Tube LAMP Using (HNB) Colorimetric Dye

Closed-tube LAMP reactions were carried out for clinical specimens and isolates using the MAST ISOPLEX^®^ DNA Lyo Kit (Mast Group, Merseyside, Liverpool, UK), with minor modifications to the manufacturer’s instructions. Lyophilized pellets were resuspended by adding 20 µL reconstitution buffer, 58 µL molecular grade water, and 2 µL of colorimetric dye (HNB). A set of six types of primers was enrolled, as in [9,19]. The reaction mixture was prepared in a total volume of 10 µL, accounting for 8 µL of LAMP reaction, 1 µL of primer mix that included 20 pmol of each FIP and BIP primer, 2.5 pmol of each F3 and B3 primers, 10 pmol of each LF and LB primers for each gene (*mecA* and *spa*), and 1 µL of extracted DNA. A positive and negative control of the kit were prepared as per the manufacturer’s instructions. LAMP reaction mixtures were incubated in a Heat Block Thermo-Shaker (Grant Bio, Cambridge, UK) for 40 min at 64 °C. A visual sky-blue color indicates the presence of the gene. In comparison, a purple color designates the absence of the gene. 

### 2.5. Detection Limit of Closed-Tube LAMP Using HNB Colorimetric Dye

The detection limit of closed-tube LAMP using HNB colorimetric dye assay was performed using 10-fold serial dilutions of genomic DNA and bacterial suspension of confirmed MRSA from clinical isolates and specimens. DNA concentrations were evaluated using NanoDrop™ Lite Spectrophotometer (ThermoFisher Scientific, New York, NY, USA).

### 2.6. Conventional Polymerase Chain Reaction (PCR) 

PCR reactions were carried out using the HotstarTaq plus master mix kit (Qiagen, Hilden, Germany). The reaction mix was prepared as 25 µL of HotStarTaq Master Mix, 7 µL of RNase-free water, 5 µL of genomic DNA, and 1 µL (2 µM) of each primer, *mecA,* and *spa* genes. The primers were as follows: *MecA* forward primer: 5′AAAATCGATGGTAAAGGTTGGC 3′; *mecA* reverse primer: 5′ AGTTCTGGAGTACCGGATTTGC 3′; *spa* forward primer: 5′ TAAAGACGATCCTTCGGT GAGC 3′; and *spa* reverse primer: 5′ CAGCAGTAGTGCCGTTTGCTT 3′ [26]. The reaction was amplified using Biometra TAdvanced thermal cycler (Biometra, Göttingen, Germany) under the following conditions: initial denaturation at 95 °C for 10 min. This is followed by 40 amplification cycles for *mecA* gene and 35 cycles for *spa* gene, consisting of denaturation at 95 °C for 30 s, annealing at 53 °C (*mecA* gene) and 50 °C *(spa* gene) for 30 s, and extension at 72 °C for 1 min. Then, the final extension is at 72 °C for 10 min. After that, 5 µL of PCR-amplified products were subjected to electrophoresis in 1.2% agarose (Agarose- LE, Ambion^®^, New York, NY, USA) and visualized using iBright™ CL1000 Imaging System (Thermo Fisher, New York, NY, USA). 

### 2.7. Data Analysis 

Data presented as specificity, sensitivity, positive predictive value (PPV), negative predictive value (NPV), and Cohen’s Kappa were calculated using equations listed in (Table 1) [27]. Calculations were based on conventional PCR as the gold standard [28,29]. Sensitivity, which measures the correctly identified positive portion, is the true positive. On the other hand, specificity measures the correctly identified negative portion, described as the true negative [30]. Cohen’s Kappa (CK) statistics measure inter-rater agreement for categorical items to check the test reliability. According to CK interpretation [31], the agreement level is none if Kappa value ranges between 0 and 0.20; minimal, if Kappa value is 0.21–0.39; weak, if Kappa value is 0.40–0.59; moderate, if Kappa value is 0.60–0.79; strong, if Kappa value is 0.80–0.9; and almost perfect, if the Kappa value is above 0.90. The positive predictive value (PPV) is the proportion of bacterial strains giving positive resistant test results, which are genuinely resistant. In contrast, the negative predictive value (NPV) is the proportion of bacterial strains showing negative resistant test results, which are true positive as measured by conventional PCR [30].

## 3. Results

### 3.1. Clinical Isolates

#### 3.1.1. Clinical Isolates Identification

According to HMC identification methods, of 218 samples, 125 clinical isolates were identified as 101 MRSA and 24 MSSA.

#### 3.1.2. Cefoxitin Disk Diffusion Method (Kirby–Bauer Test) 

One hundred twenty-five *S. aureus* clinical isolates were subjected to the disk diffusion assay. The method revealed 101 as MRSA isolates with zones of inhibitions (≤21 mm) (Figure 1A), and 24 MSSA with inhibition zones of (≥22mm) (Figure 1B). The assay specificity and sensitivity were 100%, and CK was 1.

#### 3.1.3. Closed-Tube LAMP Using (HNB) Colorimetric Dye

MRSA and MSSA clinical isolates were processed using a closed-tube LAMP (HNB) colorimetric dye, based on the visual change of the color. If the clinical specimen is positive for *mecA* or *spa*, the color will change from violet to blue (Figure 2). A total of 98 *S. aureus* isolates were identified as MRSA and 24 as MSSA. However, three were identified as false negatives, with 100% and 97% specificity and sensitivity, respectively, and Cohen’s Kappa of 0.926. 

#### 3.1.4. Conventional PCR 

In total, 101 MRSA and 24 MSSA were determined using conventional PCR. Based on the amplification products, the band cut-off size of 533 bp indicates *mecA* gene presence, and 180–670 bp denotes *spa* gene presence (Figure 3). 

#### 3.1.5. Detection limit of Closed-Tube LAMP Using HNB Colorimetric Dye 

The closed-tube LAMP incorporating HNB colorimetric dye approach was able to detect a minimum of 0.0046 ng/µL MRSA DNA and 1.5 × $10^{7}$ CFU/mL MRSA bacterial cells.

### 3.2. Clinical specimens 

#### 3.2.1. Clinical Specimen Identification 

Based on HMC identification methods, of the 218 samples, 93 were clinical specimens encompassing 57 MRSA and 36 MSSA.

#### 3.2.2. Closed-tube LAMP Results Using HNB Colorimetric Dye on Clinical Specimens 

A total of 93 clinical specimens were tested with closed-tube LAMP using HNB colorimetric dye assay to detect the presence of mecA gene and the *spa* gene. In total, 52 out of 57 specimens were identified as MRSA (Figure 4A). Furthermore, 36 out of 36 specimens were correctly identified as MSSA (Figure 4B). However, five samples were identified as false negatives. The assay revealed 100% specificity, 91.23% sensitivity, and 0.90 CK. 

All in all, specimens were divided into two categories: liquid or tissue, including placenta, knee, pus, bone, and ulcer tissues. After applying this grouping, specificity, sensitivity, and CK were computed as listed in (Table 2). 

### 3.3. Comparison of All Identification Methods and Sample Types

Our study showed that the cefoxitin disk diffusion test has a specificity of 100%, sensitivity of 100%, CK score of 1.00, and PPV and NPV were 100% compared to conventional PCR. While closed-tube LAMP using HNB colorimetric dye assay had a specificity of 100%, sensitivity of 97%, CK score of 0.926, PPV of 100%, and NPV of 88.89%. On the other hand, clinical specimens showed a specificity of 100%, a sensitivity of 91.23%, CK score of 0.90, PPV of 100%, and NPV of 87.8%, as listed in (Table 3).

## 4. Discussion 

MRSA has been reported to cause more infections than any other multi-drug resistant (MDR) Gram-negative bacteria, owing to its widespread prevalence in community and hospital settings [32,33,34,35,36]. Therefore, designing a proper choice of treatment plans, implementing preventive measures, and establishing a quick, efficient, and cost-effective diagnostic test are critical for managing the spread of MRSA. The Cefoxitin disk diffusion test is routinely used to diagnose MRSA because this technique is cost-effective and requires limited reagents available in any conventional microbiology laboratory [14,37,38]. In this study, enrolment of cefoxitin disk diffusion in MRSA detection yielded 100% sensitivity, 100% specificity, 1 CK score, compared to conventional PCR as a gold standard. However, cefoxitin disk diffusion requires an 18–24 h incubation period and an additional 14–24 h for isolation of the suspected organism, which limits its functionality in the emergency department (ED) or intensive care unit (ICU), where the immediate need for identification with a shorter time is vital [39]. Alternatively, the closed-tube LAMP using HNB colorimetric dye assay allows results to be analyzed visually, which speeds up the turnaround time to 45 min. It does not require lengthy incubation steps during detection [40]. Accordingly, it will aid in reducing the length of patient’s stay in hospital and excess economic burden [41]. In addition, the strength of the LAMP assay comes from the three designed sets of primers, forward, backward, and loop, that can detect up to eight locations on the DNA template. Moreover, the LAMP lyophilized pellet can be maintained for longer at room temperature, eliminating the necessity for sophisticated deep-freezing devices and thus reducing the overall need for complicated instruments in the method execution. Another distinctive characteristic of the closed-tube LAMP assay is its association with less contamination [42,43], thus inducing few false negatives while identifying the organism.

Another MRSA and MSSA identification method used in diagnostic laboratories is the MALDI-TOF test. Hulme J. et al., (2017) revealed that MALDI-TOF can detect a minimum of $1\times 10^{5}$ CFU MRSA bacterial cells [44]. In this study, closed-tube LAMP using (HNB) colorimetric dye detected a minimum of 1.5 × 10⁷ CFU of suspected bacterial cells and a minimum of 0.0046 ng/µL of bacterial DNA concentration. Considering the detection limit variation between MALDI-TOF and closed-tube LAMP using (HNB) colorimetric dye, MALDI-TOF needs an extra 16-18 h to obtain pure colonies in addition to the high cost of the machine, whereas as revealed by our study of closed-tube LAMP using (HNB) colorimetric dye, it takes less than 1 h, and there is no need for high-cost machines. 

PCR-based MRSA and MSSA detection techniques has shown high specificity and sensitivity [45]. However, a study by Lim et al. (2013) found that LAMP-based assays are five times more sensitive than PCR-based assays [46]. Moreover, according to Anupama et al., after comparing conventional PCR and RT-PCR against the LAMP assay, the study found that LAMP is more sensitive and specific than conventional PCR [47]. Yet, the same study concluded that RT-PCR has higher sensitivity than LAMP using HNB colorimetric dye. However, it requires an expensive instrument.

Furthermore, it was discovered in another study that PCR cannot detect low concentrations of bacteria without an enrichment step; it can spot a minimum of 12.5 ng/µL [16,44]. A closed-tube LAMP using HNB colorimetric dye detected a minimum of 0.0046 ng/µL bacterial DNA and 1.5 × 10⁷ CFU bacterial cells, as revealed by our study. According to Khosravi et al. [45], using a Multiplex PCR assay shortened the time to approximately two hours; nevertheless, results need several primer sets.

Moreover, incorporating colorimetric dye (HNB) facilitates MRSA detection by visualizing the color change based on the gene’s presence or absence. Previous studies also enrolled LAMP-based assays in microorganism identification, including COVID-19 [18,40,48], Hepatitis B Virus (HBV) [49], Brucella [43], and MRSA [45,50]. However, very few worked on closed-tube LAMP using HNB colorimetric dye and lyophilized pellets for bacterial detection from the clinical specimen using a heat block. Overall, upon performing closed-tube LAMP using HNB colorimetric dye, our study exhibited a specificity of 100%, a sensitivity of 91.23%, and CK score 0.90 among clinical specimens. Correspondingly, using the same technique demonstrated a specificity of 100%, a sensitivity of 97%, and CK scores 0.96 in detecting MRSA and MSSA among clinical isolates. Yet, among liquid clinical specimen categories, although a strong agreement was observed with the HGH findings with CK score of 0.8 and specificity of 100%, sensitivity was lowered when detecting MRSA from clinical blood specimens to 81.81% and clinical wound swab specimens to 85.71%. This is possibly due to LAMP inhibition factors in blood and wound swabs, which need more optimization in further studies. Changing the DNA extraction method to a kit or other traditional DNA extraction methods may produce more effective results in future investigations.

## 5. Conclusions

In conclusion, using the colorimetric dye HNB assay, our closed-tube LAMP demonstrates dependability with notable high sensitivity and specificity and a quick turnaround time of 45 min without expensive equipment. It can be applied directly to clinical specimens without the need for cultivation. Because of its ability to expedite the identification of isolated organisms from critically ill patients and facilitate timely prescription of the appropriate antibiotic, this technique is highly recommended for implementation in healthcare facilities. 

## Figures and Tables

**Figure 1 microorganisms-12-00157-f001:**
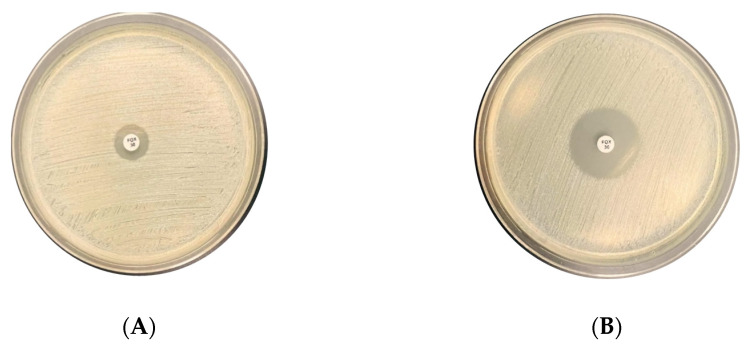
Illustration of the cefoxitin disk diffusion method in two clinical isolates. Isolate (**A**) MRSA, resistant to 30 µg Cefoxitin disk with an inhibition zone of (=16 mm). Isolate (**B**) MSSA, susceptible to 30 µg Cefoxitin disk with an inhibition zone of (>16 mm).

**Figure 2 microorganisms-12-00157-f002:**
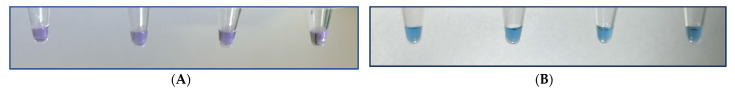
Visual color detection of closed-tube LAMP amplification products using colorimetric dye HNB among clinical isolates. (**A**) Absence of the *mecA* gene. (**B**) Presence of *mecA* gene.

**Figure 3 microorganisms-12-00157-f003:**
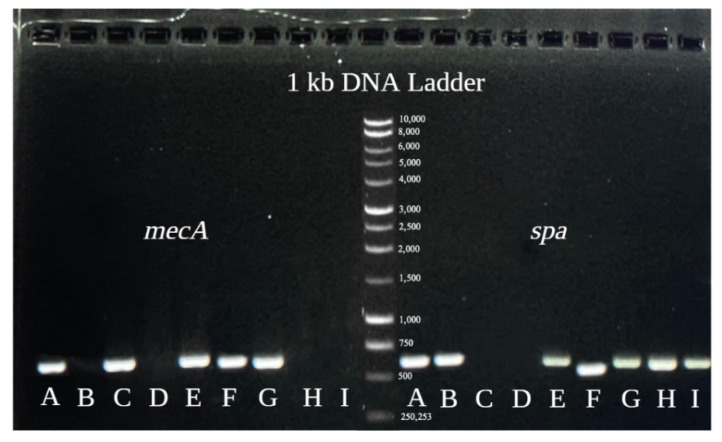
Displays of conventional PCR results. Samples to the left of the ladder correspond to *mecA* gene detection, and those to the right correspond to *spa* gene detection. Sample (A). MRSA ATCC BAA-976. Sample (B). MSSA S15. Sample (C). *S. epidermidis* ATCC 12228. Sample (D). *E. coli* ATCC 25922. Samples (E–G). Representatives of MRSA clinical isolates. Samples (H,I). Representatives of MSSA clinical isolates.

**Figure 4 microorganisms-12-00157-f004:**
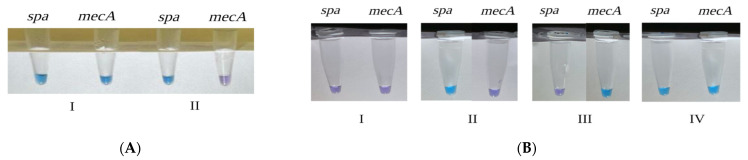
Visual detection of closed-tube LAMP using colorimetric dye (HNB) technique in two clinical specimens. (**A**): (I) MRSA (presence for *spa* and *mecA* genes); (II) MSSA (presence of *spa* gene, absence of *mecA* gene) and study controls. (**B**): (I–IV): I: *E. coli* ATCC 25922 (absence of *spa* and *mecA* genes); II: MSSA S15 (presence of *spa* gene, lack of *mecA*); III: *S. epidermids* ATCC 12228 (absence of *spa* gene, presence of *mecA* gene); IV: MRSA ATC BAA-976 (presence of both *spa* and *mecA* genes).

**Table 1 microorganisms-12-00157-t001:** Specificity and sensitivity testing criteria equations.

Testing Measurements	Equation
Specificity	$\frac{True\;Negative}{\left(True\;Negative\;+\;False\;Positive\right)}\times 100$
Sensitivity	$\frac{True\;Positive}{\left(True\;Positive\;+\;False\;Negative\right)}\times 100$
Positive Predictive Value (PPV)	$\frac{True\;Positive}{\left(True\;Positive\;+\;False\;Positive\right)}\times 100$
Negative Predictive Value (NPV)	$\frac{True\;Negative}{\left(True\;Negative\;+\;False\;Negative\right)}\times 100$
Cohen’s Kappa	$\frac{pₒ*-pₑ**}{1-pₑ}$

* pₒ: actual observed agreement; ** pₑ: chance agreement.

**Table 2 microorganisms-12-00157-t002:** Sensitivity, specificity, and Cohen’s Kappa values of different clinical specimens compared to the gold-standard conventional PCR using *mecA* and *spa*-specific primers.

Categories	Specimen Type	Total Specimens Number out of 93	Specificity	Sensitivity	CK	PPV	NPV
Liquid	Blood	22	100%	81.81%	0.8	100%	91.43%
	Urine	4	100%	100%	1		
	ETT	14	100%	100%	1		
	Sputum	6	100%	100%	1		
	Cyst fluid	1	100%	100%	1		
	Ascitic fluid	1	100%	100%	1		
	Joint fluid	1	100%	100%	1		
	Swab	3	100%	100%	1		
	Pus swab	11	100%	100%	1		
	Abscess swab	2	100%	100%	1		
	Wound swab	12	100%	85.71%	0.82		
	Drain swab	1	100%	100%	1		
	Episiotomy swab	2	100%	100%	1		
Tissue	Tissue	13	100%	75%	0.7	100%	71.43%

**Table 3 microorganisms-12-00157-t003:** Comparison of MRSA/MSSA detection methods based on sample type, specificity, sensitivity, Cohen kappa, and turnaround time.

Method	Detection	Sample Type	Specificity	Sensitivity	Cohen Kappa ^1^	PPV ^2^	NPV ^3^	Time (h)
Closed-tube LAMP using HNB colorimetric dye	*mecA* and *spa* genes	Clinical specimens	100%	91.23%	0.90	100%	87.8%	<1 h
Cefoxitin Disk diffusion test	≥22 mm MSSA ≤21 mm MRSA	Clinical isolates	100%	100%	1.00	100%	100%	≥24 h
Closed-tube LAMP using (HNB) colorimetric dye	*mecA* and *spa* genes		100%	97%	0.926	100%	88.89%	<1 h
Conventional PCR	*mecA* and *spa* genes							<4 h

^1^ Statistic measure inter-rater agreement for categorical items to check the test reliability; ^2^ positive predictive value; ^3^ negative predictive value.

## Data Availability

All data are contained within the article.

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
