# Peer review of "Rapid Visual Detection of Methicillin-Resistant Staphylococcus aureus in Human Clinical Samples via Closed LAMP Assay Targeting mecA and spa Genes"

_microorganisms, 2024, doi:10.3390/microorganisms12010157_

Round 1

Reviewer 1 Report

Comments and Suggestions for Authors

The work of the authors is about detecting MRSA and MSSA  bacteria being a huge problem internationally for patients. The work is novel and very important for hospitals. There are only minor points to address in this work prior to acceptance namely:

11.      Page 2 line 59 reference missing.

22.      Page 2 line 64 reference missing.

33.      Page 2 line 68 reference missing.

44.      Page 2 line 77 reference missing.

55.      Page 2 line 81 reference missing.

66.      Origin of the quality control organisms needs to be specified.

77.      Page 8 line 285 two points.

88.      Amount of cycles for PCR not stated.

99.      Full name of HNB dye needs to be stated.

110.   Origin of heat block thermos shaker needs to be stated in section 2.4

111.   The authors could state that especially implants need to be free of bacteria. Some groups even propose self-disinfecting implants, although they propose different approaches / different antibiotics, while metal ions might make it harder for bacteria to develop resistance.1,2

References

(1)        Kozelskaya, A. I.; Verzunova, K. N.; Akimchenko, I. O.; Frueh, J.; Petrov, V. I.; Slepchenko, G. B.; Bakina, O. V.; Lerner, M. I.; Brizhan, L. K.; Davydov, D. V.; et al. Antibacterial Calcium Phosphate Coatings for Biomedical Applications Fabricated via Micro-Arc Oxidation. Biomimetics 2023, 8 (5), 444. https://doi.org/10.3390/biomimetics8050444.

(2)        Mordovina, E. A.; Plastun, V. O.; Abdurashitov, A. S.; Proshin, P. I.; Raikova, S. V; Bratashov, D. N.; Inozemtseva, O. A.; Goryacheva, I. Y.; Sukhorukov, G. B.; Sindeeva, O. A. “Smart” Polylactic Acid Films with Ceftriaxone Loaded Microchamber Arrays for Personalized Antibiotic Therapy. Pharmaceutics 2021, 14 (1), 42. https://doi.org/10.3390/pharmaceutics14010042.

Author Response

Dear Editors:

Thank you for the thoughtful and constructive comments concerning our manuscript entitled: “Rapid Visual Detection of Methicillin-Resistant Staphylococcus aureus in Human Clinical Samples via Closed LAMP Assay Targeting mecA and spa Genes”.  The comments of the reviewers helped provide a clearer presentation and discussion of the findings of this study.  Specific responses to comments have been included below and modifications in the manuscript were highlighted in yellow. We believe that we have addressed all reviewers’ concerns and we look forward to accepting the paper for publication in the respected journal.

Reviewer 1

Comments and Suggestions for Authors

The work of the authors is about detecting MRSA and MSSA bacteria being a huge problem internationally for patients. The work is novel and very important for hospitals. There are only minor points to address in this work prior to acceptance namely:

We appreciate the valuable comments and the positive feedback.  

point 1:    Page 2 line 59 reference missing.

Respond to point 1: Reference has been added.

point 2.      Page 2 line 64 reference missing.

Respond to point 2: Reference has been added.

point 3.      Page 2 line 68 reference missing.

Respond to point 3: Reference has been added.

point 4.      Page 2 line 77 reference missing.

Respond to point 4: Reference has been added.

point 5.      Page 2 line 81 reference missing.

Respond to point 5: Reference has been added.

point 6.      Origin of the quality control organisms needs to be specified.

Respond to point 6: Thank you for your valuable comment. This part has been modified by adding this sentence “All ATCC control strains were collected from (American Type Culture Collection, US), and MSSA S15 is identified strain from Hamad Medical Corporation.” on page 3, lines 122-124. also highlighted in the text.

point 7.      Page 8 line 285 two points.

Respond to point 7: Thank you for your notice. The sentence was modified.

point 8.      Amount of cycles for PCR not stated.

Respond to point 8: The PCR cycle number and conditions were stated on page 4, lines 168-169. The sentences are highlighted in yellow.

point 9.      Full name of HNB dye needs to be stated.

Respond to point 9: The full name was mentioned on page 2, line 92 where it was first mentioned. the sentence is highlighted in yellow.

point 10.   Origin of heat block thermos shaker needs to be stated in section 2.4

Respond to point 10: thank you for your notice, this part was modified on page 4, line 152.

point 11.   The authors could state that especially implants need to be free of bacteria. Some groups even propose self-disinfecting implants, although they propose different approaches / different antibiotics, while metal ions might make it harder for bacteria to develop resistance. 1,2

Respond to point 11: Thank you for the suggestion. However, in this study, we stated one of the rapid detection methodology of Staphylococcus aureus directly from a clinical sample, the references below demonstrate an antibacterial biomaterial in treating this infection, which is not relevant. 

References

(1)        Kozelskaya, A. I.; Verzunova, K. N.; Akimchenko, I. O.; Frueh, J.; Petrov, V. I.; Slepchenko, G. B.; Bakina, O. V.; Lerner, M. I.; Brizhan, L. K.; Davydov, D. V.; et al. Antibacterial Calcium Phosphate Coatings for Biomedical Applications Fabricated via Micro-Arc Oxidation. Biomimetics 2023, 8 (5), 444. https://doi.org/10.3390/biomimetics8050444.

(2)        Mordovina, E. A.; Plastun, V. O.; Abdurashitov, A. S.; Proshin, P. I.; Raikova, S. V; Bratashov, D. N.; Inozemtseva, O. A.; Goryacheva, I. Y.; Sukhorukov, G. B.; Sindeeva, O. A. “Smart” Polylactic Acid Films with Ceftriaxone Loaded Microchamber Arrays for Personalized Antibiotic Therapy. Pharmaceutics 2021, 14 (1), 42. https://doi.org/10.3390/pharmaceutics14010042

Reviewer 2 Report

Comments and Suggestions for Authors

I have reviewed the manuscript "Rapid Visual Detection of Methicillin-Resistant Staphylococcus aureus in Human Clinical Samples via Closed LAMP Assay Targeting mecA and spa Genes." The review article presents relevant information about the development of an efficient approach to rapidly screen MRSA directly from clinical specimens using a closed-tube loop-mediated isothermal amplification (LAMP) method incorporating Hydroxy-naphthol blue (HNB) colorimetric dye assay based on the presence of mecA and spa genes. This manuscript is engaging, well-written, and adequately described. I consider this manuscript can be published after some minor changes: 1. Write the scientific names of bacteria in italics and 2. Explain clearly the difference between clinical specimens and bacterial isolates.

Author Response

Thank you for the thoughtful and constructive comments concerning our manuscript entitled: “Rapid Visual Detection of Methicillin-Resistant Staphylococcus aureus in Human Clinical Samples via Closed LAMP Assay Targeting mecA and spa Genes”.  The comments of the reviewers helped provide a clearer presentation and discussion of the findings of this study.  Specific responses to comments have been included below and modifications in the manuscript were highlighted in yellow. We believe that we have addressed all reviewers’ concerns and we look forward to accepting the paper for publication in the respected journal.

Reviewer 2

I have reviewed the manuscript "Rapid Visual Detection of Methicillin-Resistant Staphylococcus aureus in Human Clinical Samples via Closed LAMP Assay Targeting mecA and spa Genes." The review article presents relevant information about the development of an efficient approach to rapidly screen MRSA directly from clinical specimens using a closed-tube loop-mediated isothermal amplification (LAMP) method incorporating Hydroxy-naphthol blue (HNB) colorimetric dye assay based on the presence of mecA and spa genes. This manuscript is engaging, well-written, and adequately described. I consider this manuscript can be published after some minor changes:

point 1. Write the scientific names of bacteria in italics and

Respond to point 1: Thank you for this notice. All the bacteria’s scientific names have been revised and amended.

point 2. Explain clearly the difference between clinical specimens and bacterial isolates.

Respond to point 2: This part has been clarified on page 3, lines 104-107. The modified sentence is highlighted in yellow.

Reviewer 3 Report

Comments and Suggestions for Authors

The article was submitted for review - Rapid Visual Detection of Methicillin-Resistant Staphylococcus aureus in Human Clinical Samples via Closed LAMP Assay Targeting mecA and spa Genes.

The emergence of antimicrobial resistance (AMR), particularly methicillin-resistant Staphylococcus aureus (MRSA), poses a significant global health threat as these bacteria become increasingly resistant to the most available therapeutic options. Developing an effective approach for rapid screening of MRSA directly from clinical specimens becomes vital. The authors of the paper developed a closed-tube isothermal amplification method (LAMP) incorporating a colorimetric hydroxynaphthol blue assay for the direct detection of MRSA in clinical samples based on the presence of the mecA and spa genes previously identified in S. aureus. The assay demonstrated 100% specificity, 91.23% sensitivity, 0.90 Cohen's kappa (CK), 100% PPV and 87.8% NPV for clinical samples, while clinical isolates demonstrated 100% specificity, 97% sensitivity. 0.926 CK, 100% PPV, and 88.89% NPV. The study showed that LAMP sealed tube (HNB) dye is a rapid method with a turnaround time of less than 1 hour and high specificity and sensitivity.

The article is highly relevant, meaningful, and is of great importance for practice. The abstract is very well written and reflects the content of the article. The introduction substantiates the relevance of the study. Modern methods of detection and processing of material were used - polymerase chain reaction, Cefoxitin Disk Diffusion Method, Closed-tube LAMP using (HNB) Colorimetric Dye, Detection Limit of Closed-Tube LAMP using HNB Colorimetric Dye. Adequate methods of statistical data processing were also used. The results of the study are very well presented in the form of images, drawings, photos. In conclusion, using the colorimetric HNB dye assay, our LAMP372 closed tube demonstrates reliability with noticeably high sensitivity. The discussion is well presented and the results are critically and comparatively analyzed. For this purpose, 43 literary sources were used, including recent ones. In conclusion, it is said that due to its ability to accelerate, the identification of isolated microorganisms from critically ill patients contributes to the timely prescription of the appropriate antibiotic. The technique is highly recommended for implementation in healthcare.

The article may be submitted for publication without changes.

Author Response

The article was submitted for review - Rapid Visual Detection of Methicillin-Resistant Staphylococcus aureus in Human Clinical Samples via Closed LAMP Assay Targeting mecA and spa Genes.

The emergence of antimicrobial resistance (AMR), particularly methicillin-resistant Staphylococcus aureus (MRSA), poses a significant global health threat as these bacteria become increasingly resistant to the most available therapeutic options. Developing an effective approach for rapid screening of MRSA directly from clinical specimens becomes vital. The authors of the paper developed a closed-tube isothermal amplification method (LAMP) incorporating a colorimetric hydroxynaphthol blue assay for the direct detection of MRSA in clinical samples based on the presence of the mecA and spa genes previously identified in S. aureus. The assay demonstrated 100% specificity, 91.23% sensitivity, 0.90 Cohen's kappa (CK), 100% PPV and 87.8% NPV for clinical samples, while clinical isolates demonstrated 100% specificity, 97% sensitivity. 0.926 CK, 100% PPV, and 88.89% NPV. The study showed that LAMP sealed tube (HNB) dye is a rapid method with a turnaround time of less than 1 hour and high specificity and sensitivity.

The article is highly relevant, meaningful, and is of great importance for practice. The abstract is very well written and reflects the content of the article. The introduction substantiates the relevance of the study. Modern methods of detection and processing of material were used - polymerase chain reaction, Cefoxitin Disk Diffusion Method, Closed-tube LAMP using (HNB) Colorimetric Dye, Detection Limit of Closed-Tube LAMP using HNB Colorimetric Dye. Adequate methods of statistical data processing were also used. The results of the study are very well presented in the form of images, drawings, photos. In conclusion, using the colorimetric HNB dye assay, our LAMP372 closed tube demonstrates reliability with noticeably high sensitivity. The discussion is well presented and the results are critically and comparatively analyzed. For this purpose, 43 literary sources were used, including recent ones. In conclusion, it is said that due to its ability to accelerate, the identification of isolated microorganisms from critically ill patients contributes to the timely prescription of the appropriate antibiotic. The technique is highly recommended for implementation in healthcare.

The article may be submitted for publication without changes.

Respond to the reviewer: Thank you for the positive feedback on our manuscript.